# Theoretical and Experimental Study of an Electrokinetic Micromanipulator for Biological Applications

**DOI:** 10.3390/biomimetics10010056

**Published:** 2025-01-15

**Authors:** Reza Hadjiaghaie Vafaie, Ali Fardi-Ilkhchy, Sobhan Sheykhivand, Sebelan Danishvar

**Affiliations:** 1Department of Electrical Engineering, University of Bonab, Bonab 5551761167, Iran; 2Department of Material Engineering, University of Bonab, Bonab 5551761167, Iran; 3Department of Biomedical Engineering, University of Bonab, Bonab 5551761167, Iran; sheykhivand@ubonab.ac.ir; 4College of Engineering, Design, and Physical Sciences, Brunel University London, Uxbridge UB8 3PH, UK

**Keywords:** microfluidic chip, MEMS, biological solution, electrokinetics, micromanipulation, concentrator

## Abstract

The ability to control and manipulate biological fluids within microchannels is a fundamental challenge in biological diagnosis and pharmaceutical analyses, particularly when buffers with very high ionic strength are used. In this study, we investigate the numerical and experimental study of fluidic biochips driven by ac electrothermal flow for controlling and manipulating biological samples inside a microchannel, e.g., for fluid-driven and manipulation purposes such as concentrating and mixing. By appropriately switching the voltage on the electrode structures and inducing AC electrothermal forces within the channel, a fluidic network with pumping and manipulation capabilities can be achieved, enabling the control of fluid velocity/direction and also fluid rotation. By using finite element analysis, coupled physics of electrical, thermal, fluidic fields, and molecular diffusion transport were solved. AC electrothermal flow was studied for pumping and mixing applications, and the optimal model was extracted. The microfluidic chip was fabricated using two processes: electrode structure development on the chip and silicon mold fabrication in a cleanroom. PDMS was prepared as the microchannel material and bonded to the electrode structure. After implementing the chip holder and excitation circuit, a biological buffer with varying ionic strengths (0.2, 0.4, and 0.6 [S/m]) was prepared, mixed with fluorescent particles, and loaded into the microfluidic chip. Experimental results demonstrated the efficiency of the proposed chip for biological applications, showing that stronger flows were generated with increasing fluid conductivity and excitation voltage. The system behavior was characterized using an impedance analyzer. Frequency response analysis revealed that for a solution with an electrical conductivity of 0.6 [S/m], the fluid velocity remained almost constant within a frequency range of 100 kHz to 10 MHz. Overall, the experimental results showed good agreement with the simulation outcomes.

## 1. Introduction

In recent years, miniaturization has become a critical topic in various engineering disciplines, including electronic, biomedical, biological, and fluid mechanical engineering [1]. The ability to control fluids at the micro scale is essential for performing biological analyses and processes. Fluid drive and control on the micro scale are necessary for almost all portable and lab on a chip application [2]. Due to the governing physics of microfluidics, fluid flow within channels is laminar and occurs at very low Reynolds numbers, and molecular diffusion is the dominant transport phenomenon on the micro scale. Consequently, inducing chaos and perturbation effects within microchannels poses significant challenges [3]. Implementation of these devices faces fundamental challenges, including applicability to various biological applications (typically involving high-conductivity biological buffers), low power consumption, portability, and the compatibility of different components [4]. External forces such as electromagnetic, piezoelectric, thermopneumatic, and electrokinetic have been employed for these purposes [5]. However, due to the unique advantages and characteristics of the electrokinetic approach, this study focuses on using electrohydrodynamic theory. In general, any charged particle possesses energy and generates an electric field in its surroundings [6]. A three-dimensional electrode structure that eliminates reverse flows, enabling a conveyor belt-like motion of fluid within a microchannel is introduced in [7]. In this structure, the region of the electrode that drives forward flow is enlarged, while the region causing reverse flow is minimized. For AC electrokinetic pumps, the fluid velocity depends on various factors, including the magnitude and frequency of the applied field, the electrolyte concentration, and the geometric characteristics of the channel. Other structures, such as traveling-wave electrokinetic (twEK) pumping [8,9], use traveling waves instead of asymmetric electrodes to break the symmetry. The twEK flow can be implemented in three-phase or four-phase configurations. AC electroosmotic (ACEO) methods are efficient for low-ionic-strength fluids but lose effectiveness as ionic strength increases [10].

A high-efficiency electrokinetic micromixer using four electrodes on the walls of a rectangular chamber to induce mixing is presented in [11]. Particle tracking analysis demonstrates the system’s effectiveness in generating turbulence within microchannels. Also, a zig-zag channel design was employed to achieve efficient mixing, specifically for chemical samples [12]. Micro-pumps and micromanipulators (such as micromixers and microconcentrators) have diverse applications, including DNA analyzers, protein folding systems, reactors, etc. [13]. A broad range of biological applications involves saline solutions, which have high ionic strength. As a result, electroosmotic phenomena lose their effectiveness. In most biological analyses, positive and negative ions in the solution can migrate under an electric field [14]. As shown in Figure 1, by placing two pairs of electrodes at the bottom of a microchannel and applying a high-frequency AC signal, three different frequency ranges can be discussed:(a)When the external ac electric field is slower than the fluid charging time (τq = ε/σ); since there is enough time to fully screen the electrodes by the electric field; therefore, no electric field is left in the medium.(b)When the external ac electric field is comparable to the fluid charging time (τq = ε/σ), which is shown in Figure 1A. The electrodes are partially screened by a part of the applied electric field, and the other part of the electric field falls over the fluid medium. As a result, the corresponding tangential force on the ions will cause AC electroosmotic flow in the medium.(c)When the external ac electric field is faster than the fluid charging time (τq = ε/σ) as illustrated in Figure 1B. Since there is no time to screen the electrodes, all the electric field falls over the fluid medium, but the ions do not have any role in the formation of the bulk electric field. In this frequency range, the ACEO flow is very poor, and instead, the ACET mechanism can occur due to the thermal effects inside the channel [15].

An AC electrothermal micropump with two types of asymmetric electrode structures (two-phase and single-phase) was proposed [16]. In both systems, the generated non-uniform electric field is responsible for producing thermal gradients and driving high-conductivity fluids. The electrodes were electrically actuated (up to 100 MHz), and the frequency response analysis showed that the AC electrothermal effect generated a bulk flow within the channel.

An asymmetric electrode is a proposed structure for designing an AC electrothermal micropump [17]. The effect of fluid conductivity on flow velocity is investigated, and results indicate that the flow velocity increases by increasing the medium fluid electric conductivity. For fluids with a conductivity of 0.1 S/m, the system showed no dependence on excitation frequency within the 100 kHz to 1 MHz range. Additionally, higher voltages generate stronger electrothermal forces and increase the fluid velocity. An AC electrothermal micromixer is designed with symmetric side-by-side electrode arrays to induce rotational flow [18]. Alternating positive and negative sinusoidal signals were applied to the electrodes to form electric and thermal effects that generate flows within the channel. In clinical medicine and biological tests, fluidic micromanipulators have been widely employed for the identification of biochemical products, enzyme assay, polymerization, organic synthesis, and biological screening applications [19,20,21,22]. Using the electrokinetic effect for mixing applications has many advantages, including miniaturization, no moving components, simple design, fast and efficient mixing, negligible hydrodynamic dispersion, no vibration and fatigue, and easier integration with microelectronics [23,24,25]. In this study, we propose an electrode structure to simulate and experimentally investigate pumping in various directions, as well as the creation of rotational flows inside the channel for manipulation and concentrating purposes.

## 2. Theory and Material Properties

The proposed electrode structure is designed to efficiently control and manipulate fluids by applying appropriate excitation and grouping different electrodes. As illustrated in Figure 2, the direction of fluid movement and the generation of rotational flow within the microchannel can be controlled by proper grouping of the electrodes and voltage switching.

Now, we delve into the theory of electrothermal forces within fluidic microchannels. The ACET effect describes fluid motion induced by a temperature gradient in the presence of an alternating electric field. When an electric field *E* is applied to a fluid with electrical conductivity *σ*, the phenomenon of Joule heating occurs, as described by the energy balance equation [26].(1)$$k\nabla^{2}T+\frac{1}{2}<\sigma E^{2}>=0$$

In this relationship, *T* represents temperature, and *k* denotes thermal conductivity. It is worth noting that, at this scale, thermal convection can be neglected compared to thermal diffusion [27]. When a non-uniform electric field is applied to the system, spatial variations in heat generation occur. This results in the formation of local thermal gradients within the fluid. These thermal gradients induce spatial variations in the local electrical conductivity *σ* and the local dielectric permittivity *ε* [28].(2)$$\begin{array}{c} \nabla \varepsilon =(\frac{\partial \varepsilon }{\partial T})\nabla T\\ \nabla \sigma =(\frac{\partial \sigma }{\partial T})\nabla T \end{array}$$

In addition, the gradients in *σ* and *ε* result in the generation of a mobile space charge density *ρ* within the fluid [28].(3)$$\rho =\nabla .(\varepsilon E)=\nabla \varepsilon .\vec{E}+\varepsilon \nabla .\vec{E}$$(4)$$\frac{\partial \rho }{\partial t}+\nabla .(\sigma \vec{E})=0$$

This phenomenon causes the fluid to experience a volumetric force *f_E_* [29].(5)$$f_{E}=\rho \vec{E}-\frac{1}{2}{\left|\vec{E}\right|}^{2}\nabla \varepsilon$$

This force, with the aid of the fluid’s viscosity, sets the fluid into motion. The time-averaged force exerted on the fluid can be expressed as follows, according to Equation (6) [30]:(6)$$<F_{et}>-\frac{1}{2}(\frac{\nabla \sigma }{\sigma }-\frac{\nabla \varepsilon }{\varepsilon }).\frac{\varepsilon {\left|E\right|}^{2}}{1+{\left(\omega \tau \right)}^{2}}-\frac{1}{4}\nabla \varepsilon {\left|E\right|}^{2}$$

In this relation, *τ* = *ε*/*σ* represents the charge relaxation time. For an aqueous solution at a temperature of *T* = 293 K, the following equations apply:(7)$$\frac{1}{\varepsilon }\frac{\partial \varepsilon }{\partial T}=-0.004\;\Rightarrow \;\frac{\nabla \varepsilon }{\varepsilon }=\frac{1}{\varepsilon }\frac{\partial \varepsilon }{\partial T}\nabla T=-0.004\nabla T$$(8)$$\frac{1}{\sigma }\frac{\partial \sigma }{\partial T}=0.02\;\Rightarrow \;\frac{\nabla \sigma }{\sigma }=\frac{1}{\sigma }\frac{\partial \sigma }{\partial T}\nabla T=0.02\nabla T$$(9)$$<F_{et}>=-0.012\nabla T.\frac{\varepsilon {\left|E\right|}^{2}}{1+{\left(\omega \tau \right)}^{2}}+0.001.\nabla T.\varepsilon {\left|E\right|}^{2}$$

In order to neglect the ACEO flow, the applied frequency must be large enough to disregard the charge layer formation at the fluid interface [31]. Additional concepts, including the mixing parameters, will be addressed in the next section. In general, the different characteristics of the input fluid for analyzing the various physical processes are summarized in Table 1.

## 3. Design and Simulation Study

Designing a system that is capable of controlling and manipulating fluid inside a microchannel is important. Such a system should be able to control the pumping speed and its direction while also generating rotational flows within the microchannel, which could be beneficial for mixing and concentrating operations. By considering the electrode structure, as shown in Figure 2, and applying suitable electrode excitation, the goal is to generate an ACET flow to move the fluid in various directions. Additionally, rotational flows will be induced for concentrating and mixing purposes. To calculate the AC electrothermal force in Equation (9), the electric field inside the microchannel can be solved using the Laplace equation [32]:(10)$$\nabla^{2}\varphi_{e}=0$$(11)$$\bar{E}=-\nabla \varphi_{e}$$

Given a pair of electrodes, which includes a narrow electrode and a wide electrode, electrical signals +V_0_sin(2πft) and −V_0_sin(2πf t) are applied to the electrodes, while the other boundaries are electrically insulated. Under these conditions, a non-uniform electric field is created inside the channel, and the electric field will be stronger near the narrow electrode. According to Joule’s law, applying a non-uniform electric field to a fluid with high electrical conductivity results in the generation of local heat inside the channel, which in turn alters the electrical properties of the fluid. The amount of heat and the thermal gradient created inside the fluid can be calculated using the energy equation [33]:(12)$$\rho_{m}c_{p}\bar{v}.\nabla T+\rho_{m}c_{p}\frac{\partial T}{\partial t}=k\nabla^{2}T+\sigma {\left|E\right|}^{2}$$

In this relationship, *ρ* represents the fluid density, *c_p_* is the specific heat capacity of the fluid, and *u* is the fluid velocity vector. Considering that the fluid velocity in the microchannel is very small, a good approximation is that the temperature at the input and output of the channel is equal to the ambient temperature [34]. As a result, the generated thermal gradient changes the electrical and dielectric properties of the fluid [35]. The fluid motion and the flow generated inside the microchannel can be analyzed using the continuity equation and the Navier–Stokes equations [36]:(13)$$\rho_{m}(\frac{\partial \bar{u}}{\partial t}+\bar{u}\nabla \bar{u})-\mu \nabla^{2}\bar{u}+\nabla p=f_{Bulk}$$(14)$$\nabla \bar{u}=0$$

In this context, *p* represents the pressure inside the microchannel, *μ* is the dynamic viscosity of the fluid, and *F_Bulk_* represents the bulk force applied to the fluid. The boundary conditions for pressure are described by a zero-pressure condition at the inlet and outlet and zero-pressure gradients at the walls. Given the characteristics of micro-systems, the fluid flow is entirely laminar, and the velocity at the walls follows the no-slip condition, meaning there is no velocity at the boundary [37]. In such a system, all the forces generated by ACEO, DEP, and ACET play a role in shaping the bulk force. However, because the fluid has a high electrical conductivity and the applied excitation frequency is much higher than the fluid’s natural frequency, the generated forces by ACEO and DEP can be neglected. In this case, the ACET force dominates the flow behavior inside the microchannel. Therefore, the only significant force that drives the flow is the AC electrothermal force [38], which leads to fluid motion starting from near the narrow electrode and flowing toward the wide electrode. The simulation study was carried out by considering the boundary conditions and using fluid material properties according to Table 1. By placing an array of electrodes together, we will be able to pump the fluid directed from the narrow electrode to the wide electrode (as shown in Figure 3). It can be seen from Figure 3 that a 0.2 [S/m] solution with electrical potential +/−3 V and 300 kHz is actuated, and the maximum temperature rise is about 1.5 K inside the channel (as shown in Figure 3b). The locally perturbed temperatures lead to regional temperature gradients, thus gradients in conductivity and permittivity. In combination, the interaction between the electric field, temperature field, and fluid velocity field results in an ACET flow. The maximum flow velocity occurs near the narrow electrode, where the greatest thermal gradient is created, leading to the highest electrothermal force and, thus, the highest velocity (500 μm/s above the narrow electrode). As indicated in Figure 3c, the generated flow causes an average of 60 μm/s fluid velocity at the channel outlet. It should be noted that the results in Figure 3 are simulated by using a unit cell with periodic boundary conditions (PBCs). PBCs mean a set of boundary conditions that are chosen for approximating a large (infinite) system by using a small part called a unit cell. For more visualization, we simulate the system with four units of electrode pairs, and as shown in Figure 4, the same results are achieved.

In order to simulate and design a manipulator, we studied the mixing effect by exciting the electrode in rotational mode. By dividing the center electrode into two pieces, we are able to generate forward and reverse directional effects simultaneously, which generate a circulation effect above the electrodes. Fluids A and B, with initial concentrations of 1 and 0 [mol/m^3^], respectively, were inserted inside the channel with an initial fluid velocity of 100 [μm/s] and fluid material properties according to Table 1. In order to study the mixing effect, we need to solve the mass transport equation and concentration of species inside the channel [39]:(15)$$(\bar{u}.\nabla )C-D\nabla^{2}C=0$$

In this relation, *u* represents the species velocity vector, *C* is the species concentration, and *D* is the molecular diffusion coefficient of the species. As discussed earlier, the concept of generating chaos inside the microchannel using a pair of symmetrical electrodes has been applied in [40]. One of the main challenges in AC electrothermal theory is the amount of heat generated in the system. Based on the concepts discussed for the electrothermal force, a finite element analysis was performed for a 0.2 [S/m] biological solution (with actuation parameters: +/−3V and 300 kHz). Figure 5 shows the species concentration distribution inside the three-dimensional microchannel at different heights of the channel. As long as the electrodes are not actuated, the mixing process of the species occurs solely through molecular diffusion at the interface of the two fluids (Figure 5; t = 0 s), which is a very slow phenomenon. By appropriately switching the electrodes, the direction of rotation of each unit can be changed sequentially to generate more efficient flows for the mixing process. A significant temperature rise induces a density difference in the fluid, which in turn increases the buoyancy force in the channel. The ratio of AC electrothermal force to buoyancy force is studied and concluded in [34]; if the temperature rise of the system is less than 6 Kelvin, the buoyancy force is negligible compared to the AC electrothermal force. In addition to buoyancy force, a high temperature rise can damage certain buffers and biological samples [36,41]. Therefore, it is clear that generating an effective AC electrothermal flow with minimum temperature rise is important. Reducing the amount of heat generated extends the applications of the device. A high temperature rise induces a mass density difference in fluid, and as a result, a buoyancy effect arises inside the channel. The ratio of ACET force over the buoyancy force investigated by Equation (16) [38].(16)fACETfBouyancy≈2α−βεErms2(∂ρm∂T)(gπL2)
where *g* = 9.8 m/s^2^ is the gravity acceleration, *∂ρ_m_*/*∂T*, accounts for thermal expansion of fluid, and L is the characteristic length of the system. It was proven that the ACET force is much stronger than the buoyancy force when the temperature rise is about 5 K or less [38]. In addition to the buoyancy effect, such large temperature rises can damage solutions in bio-analytical and immunoassay binding applications [41]. We generate an efficient motional effect by locally small temperature distribution inside the channel. It can be seen from the results that the maximum temperature rise for different applications can be controlled based on the applied electric potential and the fluid ionic strength.

Also, in order to further investigate the chaos mixing effect, we released two adjacent particles at the inlet of the microchannel. As shown in Figure 6, the particle trajectories illustrate that the adjacent particles will experience a bulk force, and as a result, the particles are stretched, folded, and finally separated (proof of a chaotic regime) [11,42]. The stretching, folding, and breaking up effects reveal a chaotic regime and ease the molecular diffusion transport for mixing purposes.

If complete mixing occurs, the concentration at the output will be 0.5 mol/m^3^. The mixing quality factor (Q) inside the microchannel is described by Equation (17) [43].(17)$$Q=\left[1-\frac{\int_{A}\left|c-c_{\infty }\right|d_{A}}{\int_{A}\left|c_{0}-c_{\infty }\right|d_{A}}\right]\times 100\%.$$

In this relation, *C* represents the fluid concentration, *C*_0_ represents the concentration of the fluid in the unmixed state (0 or 1 mol/m^3^), and *C_∞_* represents the concentration of the fluid in the fully mixed state (0.5 mol/m^3^). In order to study the mixing efficiency and also other impressive parameter effects, we ran our simulations with different voltages and different electric conductivities according to Equation (17). the mixing efficiency is calculated as shown in Figure 7. Meanwhile, for a 0.6 [S/m] and +/−8 V actuation voltage, mixing efficiency above 90% is achieved. It should be noted that the temperature rise in the system is 13.3 degrees Kelvin, which is harmful for some biological samples. Meanwhile, the temperature rises in the channel and decreases by decreasing the electric potential. Therefore, we suggest using low electric potential, and the high mixing efficiency will be achieved by adding another pair of electrodes.

## 4. Fabrication and Test Process

The fabrication process is carried out based on the concepts of AC electrothermal effects and the simulation results. From a fabrication point of view, we determined the appropriate electrode structure for electric actuation, the dimensions of the microchannel, and the necessary conditions for selecting the suitable substrate and microchannel. Therefore, a silicon wafer was chosen for the substrate, platinum for the electrodes, and PDMS material for the microchannel. The micromanipulator fabrication process, based on the AC electrothermal effect, consists of two process flows, as shown in Figure 8. The first process flow is for the electrode structure deposition, and the second one is for the fabrication of the Si mold for the PDMS microchannel.

### 4.1. First Process Flow

The first process flow was carried out on a 4-inch p-type silicon wafer with a thickness of 525 μm and electrical conductivity characteristics of 0.1–0.5 Ω.cm. The surface of the wafer was cleaned using the RCA method in a wet bench prior to high-temperature processes such as LPCVD and thermal oxidation. To achieve electrical isolation and thermal matching of the electrodes, a 100 nm thick silicon dioxide (SiO_2_) layer was grown on the silicon wafer through thermal oxidation. Titanium and platinum metals, with thicknesses of 20 and 200 nanometers, were deposited on the entire wafer using the sputtering method. The design of the required electrode structures, microchannel structures, and markers (for wafer cutting purposes) on the 4-inch wafer and 5-inch mask was carried out using Clewin software ver.4, as shown in Figure 9. It should be noted that different geometrical sizes are designed for ACET, the proposed microchip. The efficient sizes are selected based on Taguchi optimization results. Therefore, 16 different topologies for electrode structures are fabricated for different applications such as pumping, mixing, and concentrating. The electrical connections of the microchips are tested, and the chips are prepared for experimental investigation.

### 4.2. Second Process Flow

For the lithography process, a dehydration step was performed on the wafers, which were placed on a hot plate at 130 °C for 15 min to ensure the photoresist would adhere properly to the metallic wafer surface. After dehydration, the AZ9221 positive photoresist was coated on the wafers with a thickness of 2 μm. The etching process was performed using Ion Beam Etching (IBE), which has excellent selectivity for titanium (Ti) and platinum (Pt). After the etching process, the remaining photoresist on the electrodes was removed. Since we used a dry etching process for a better stripping process, the wafers were first treated with oxygen plasma at 1000 W for 1 min to dry-strip the top layer with plasma, and then the wafers were wet-stripped using solution 1165. After the photoresist stripping process, the electrode structures were visible on the silicon wafer, as shown in Figure 8. After completing the first process, AZ-ECI photoresist was coated on the wafer to a thickness of 5 μm to act as a protective layer for the electrodes during the chip dicing process.

### 4.3. Microchannel Fabrication

After the Si mold was completed, the fabrication of the PDMS microchannel began. This process included surface activation, mixing the PDMS solution with the catalyst, degassing, coating the PDMS material onto the molded chip, curing the PDMS, detaching the PDMS, creating the inlet/outlet openings for the channel, aligning the channel with the substrate, and finally bonding the microchannel to the substrate. Initially, the silicon mold was subjected to a silanization process for 30 min using a desiccator and 3 drops of TMCS (TriMethylChloroSilane) solution. The PDMS material and catalyst solution were mixed in a 10:1 ratio in a mixer for 1 min at 2000 rpm. Following this, the degassing process was conducted for 2 min. To ensure better degassing, the mixture was placed in a desiccator for a suitable amount of time, and then the PDMS material was coated to the Si mold (using a petri dish as a container). The container was then placed in an oven at 80 °C for 2 h for curing. After curing, the PDMS microchannel was detached from the Si mold using a blade. The inlets and outlets of the microchannel were opened using an optical punch. The microchannel, along with the IPA solution, was placed in an ultrasonic bath to remove small particles from the entrances and walls of the microchannel. Finally, the surface of the PDMS microchannel was activated using oxygen plasma with a power of 30 watts, making it easier to bond with the electrode wafer, and the microchip was prepared for testing. One of the most important requirements for biofluidic chips is biocompatibility. The PDMS microchannel and Pt electrodes are well-known biocompatible materials for most biological samples, DNA analysis, PCR, enzyme assays, and integrated analytical detections such as on-chip NMR, FRIT, and Raman spectroscopy [44,45,46,47,48]. It should be noted the parylene coating can improve the biocompatibility of a microfluidic device [49].

### 4.4. Test Process

To implement the testing process, an electronic circuit was designed, including four signal amplifiers for electrode activation. Two of the amplifiers were biased in an inverting mode, and the other two were biased in a non-inverting mode, allowing the alternating positive and negative signals to be easily applied to the electrical contacts of the fluidic chip via jumpers (as shown in Figure 10). The designed PCB circuit not only supplies the driving circuits but also includes a space for the chip holder, as shown in the figure, which is responsible for keeping the chip fixed during the testing process.

## 5. Experimental Results and Discussion

The test process was carried out using fluorescent particles. Due to the high capability of the fluorescent method in measuring and characterizing the system, this approach is generally preferred [50]. A series of tests were performed to investigate the effects of actuation parameters (frequency and voltage) as well as changes in fluid conductivity on the performance of the ACET flow. In the experimental tests, the microscope’s field of view was adjusted to a height of about 5 μm above the electrode surface. The biological buffer with an electrical conductivity range of 0.2–0.6 [S/m] was mixed with fluorescent particles and loaded into the microchannel. Upon applying the actuation signal of +/−3 V_rms_ at a frequency of 300 kHz, similar to the simulation results, the movement of the fluorescent particles from the smaller electrode to the larger electrode was observed. The non-uniform electric field and thermal gradient were identified as the driving forces for the particles’ movement in a specific direction. As shown in Figure 11a, the biological sample with an electrical conductivity of 0.2 [S/m] was mixed with fluorescent particles and loaded into the microchannel. Upon applying the actuation signal of +/−3 V_rms_ at a frequency of 300 kHz, the movement of the fluorescent particles was observed. Figure 11a illustrates the movement paths of particles over time. The average speed in this test was 70 μm/s. Figure 11b indicates an average fluid velocity of 145 μm/s for the 0.6 [S/m] solution.

According to the tests, as shown in Figure 12a, increasing the conductivity of the fluid and the applied voltage increases the flow speed. Additionally, frequency analysis of the system shows that as the conductivity of the fluid increases, the operational frequency range expands. For a buffer with an electrical conductivity of 0.2 S/m, the AC electrothermal flow speed remains constant within the frequency range of 100 kHz to 8 MHz and decreases thereafter. In contrast, for a buffer with a conductivity of 0.6 S/m, the AC electrothermal flow speed remains constant within the frequency range of 100 kHz to 10 MHz and decreases afterward. The frequency behavior of the fluid inside the microchip was also examined using an impedance analyzer. Regarding frequency, DI water, with conductivity of a few microsiemens per meter, has a magnitude in the kilo-ohm range and a phase angle between −60° and −80°, which is due to the capacitive nature of the EDL (Electrical Double Layer). This capacitive property indicates the formation of a thick EDL layer, making it very suitable for ACEO and DC-biased ACEO applications. The biological solution with an electrical conductivity of 0.2 S/m was loaded into the microchip, and impedance analysis in the frequency range of 100 kHz to 1 MHz showed a magnitude of 50 ohms and a phase angle of −9° in the system. These properties indicate the resistive nature of the system, making it suitable for AC electrothermal actuation. As the conductivity of the PBS fluid increases, the resistive nature of the system increases, leading to more efficient flow generation under the same voltage excitation and, simultaneously, more heat generated in the system. Figure 12b indicates the generated flow velocity at different frequencies; it should be noted that because of the amplifier’s frequency limitation, the tests were carried out from 100 kHz up to 140 MHz. The fluid velocity comparison between simulation results and experimental results shows that there is an agreement above 80% between the results.

For more manipulation purposes, the electrodes are also biased in rotational mode. As shown in Figure 13, the experimental result indicates the generation of rotation flow, which enhances the perturbation effect by combining the forward and reverse pumping effects together in a short time and over a short distance. If the actuation voltage of the electrodes increases more, then the perturbation mechanism will be dominant, and more significant vortices will be induced. Micromixers and sensor concentrators can be potential applications of such techniques.

Figure 13a,b shows both the simulation and experiment rotational flow inside the microfluidic channel. It needs two conditions to achieve a chaotic regime in laminar channels [51]: (a) the system has three independent dynamic variables, and (b) the equations of motion contain a nonlinear term that couples several of the variables. As shown in the simulation part for the mixing effect, the generated ACET rotational flow enhances the molecular diffusion transport by simply stretching, folding, and breaking the fluid up. In biological studies, optimization and regulation of gene expression are of great interest [52,53]. Recently, machine learning and artificial intelligence methods have been used in these studies [54,55,56,57]. The use of Artificial Intelligence methods has recently been used in the treatment of various cancers [58], including thyroid [59], breast [60], liver [61], identification of individuals [62], development of vaccines [63,64], vision control [65] and etc. In future studies, we plan to use AI [66] in combination with the present study.

## 6. Conclusions

In this study, the control and manipulation of biological solutions and buffers within a microchip were investigated both through simulations and experiments. AC electrothermal force was introduced as an efficient force compared to other electrokinetic methods for biological applications where conductive buffers and solutions were used. By adjusting the actuation voltage/frequency and properly selecting the electrodes, a microfluidic chip was designed for the purposes of pumping (controlling the speed and direction of the fluid) and analyte and buffer perturbation. AC electrothermal force has disadvantages, including power consumption and heat generation within the microchip. The heat generated is dependent on the fluid’s electrical conductivity and the actuation voltage. However, the heat produced in the system is acceptable for many purposes. The proposed chip is fabricated, and the effects of the fluid’s electrical conductivity and electric voltage/frequency on system performance were examined. Experimental results showed a good potential for fluid control and manipulation inside the channel and above 80% agreement with simulation results.

Integration of conventional detection/analytical methods (such as NMR, IR transmission, FTIR, and Raman spectroscopy) with our developed microchip will be of interest for a variety of applications such as molecular structure analysis [44,45], enzyme assay [46,47] and protein folding/refolding [48] applications.

## Figures and Tables

**Figure 1 biomimetics-10-00056-f001:**
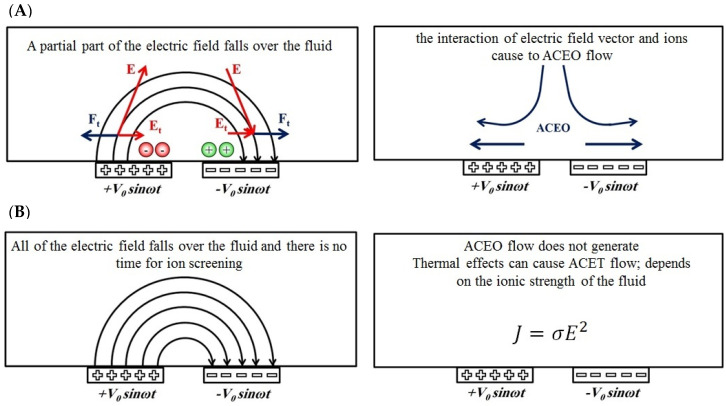
AC electric field and fluid interaction at different frequencies. (**A**) Applied electric field frequency is comparable with the fluid frequency. (**B**) Applied electric field frequency is much higher than the fluid frequency.

**Figure 2 biomimetics-10-00056-f002:**
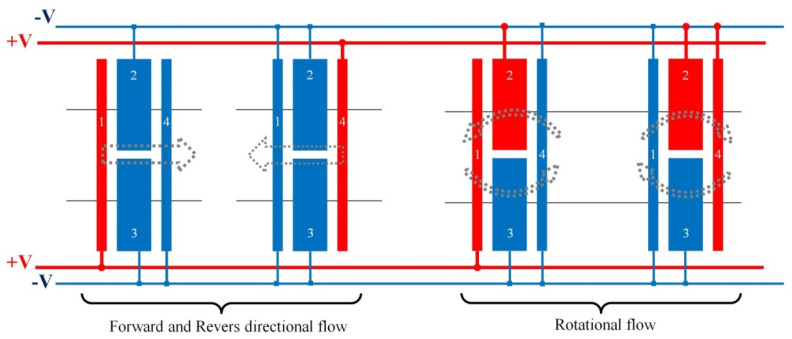
Schematic view of our electrode’s layout and pads for electrical connections for different manipulation modes.

**Figure 3 biomimetics-10-00056-f003:**
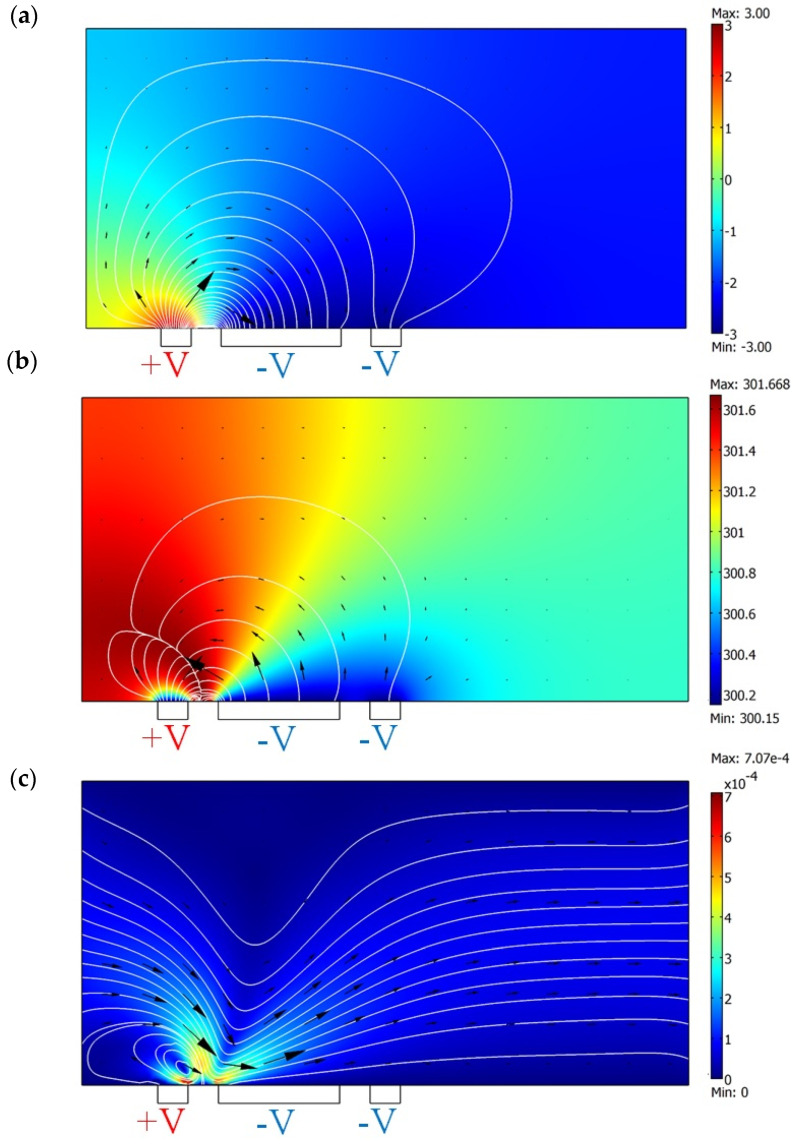
Simulation results for ACET flow. (**a**) Surface plot indicates electric potential [V], and the arrow and streamlines indicate electric field [V/m]. (**b**) Surface plot illustrates temperature [K], and the arrow and streamlines illustrate temperature gradient [K/m]. (**c**) Surface plot, arrow and streamlines depict fluid velocity field [m/s].

**Figure 4 biomimetics-10-00056-f004:**
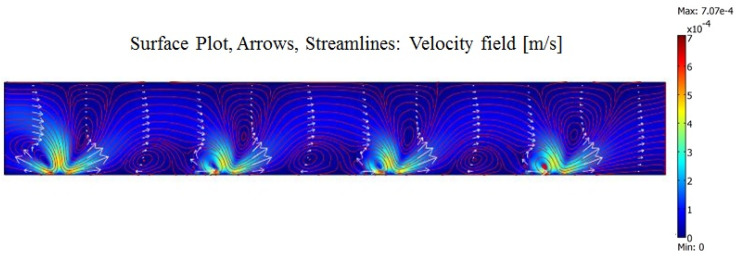
Simulation results for four units of electrodes and resulting in the generated AC electrothermal flow.

**Figure 5 biomimetics-10-00056-f005:**
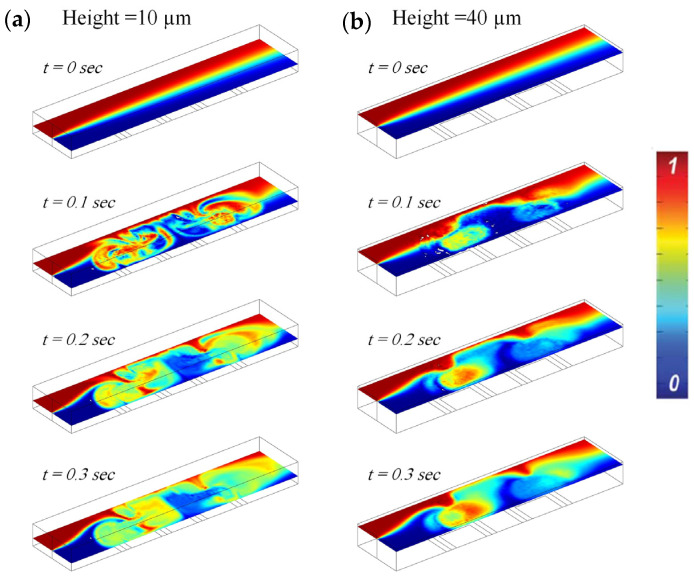
The species concentration distribution inside the microchannel for different times: (**a**) 10 μm height; (**b**) 40 μm height.

**Figure 6 biomimetics-10-00056-f006:**
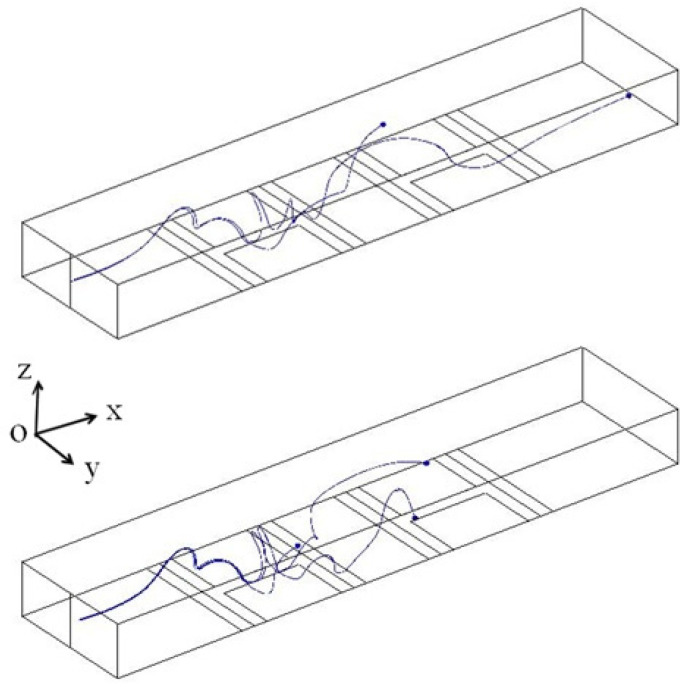
Particle trajectories and the resulting stretching and folding effects.

**Figure 7 biomimetics-10-00056-f007:**
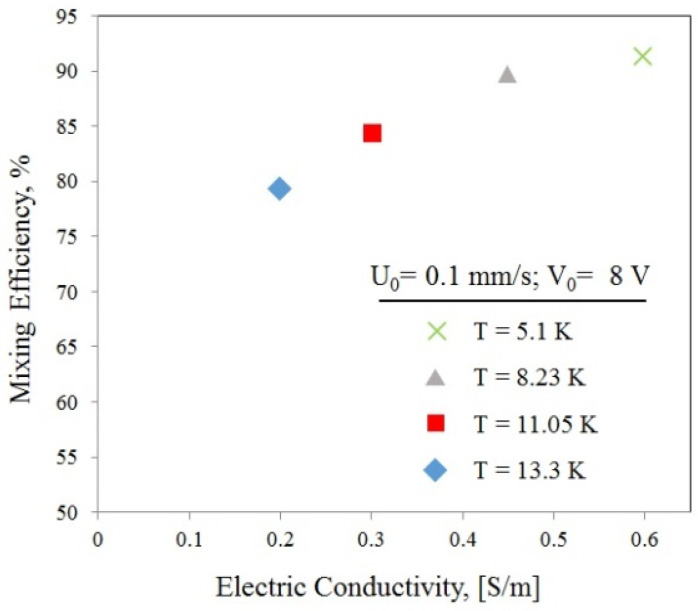
Mixing efficiency and generated temperature rise for different electric conductivities.

**Figure 8 biomimetics-10-00056-f008:**
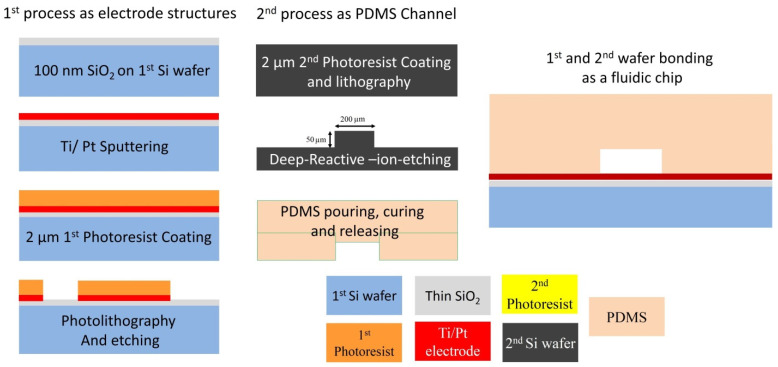
Fabrication process flow for electrode structures, silicon mold, and PDMS microchannel.

**Figure 9 biomimetics-10-00056-f009:**
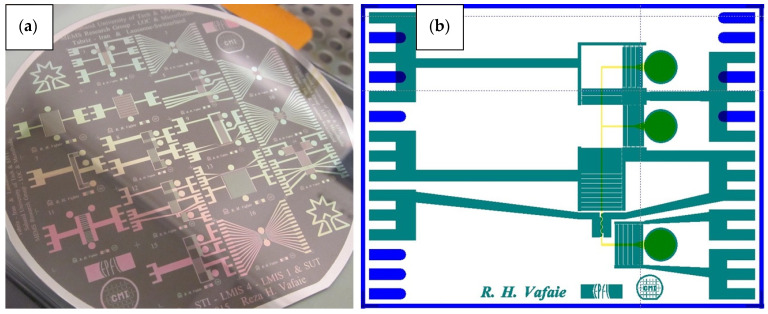
Electrode structures: (**a**) electrode structure’s wafer with different patterns for manipulation purposes (before wafer cutting); (**b**) a layout for one of the electrode chips along with the connection pads.

**Figure 10 biomimetics-10-00056-f010:**
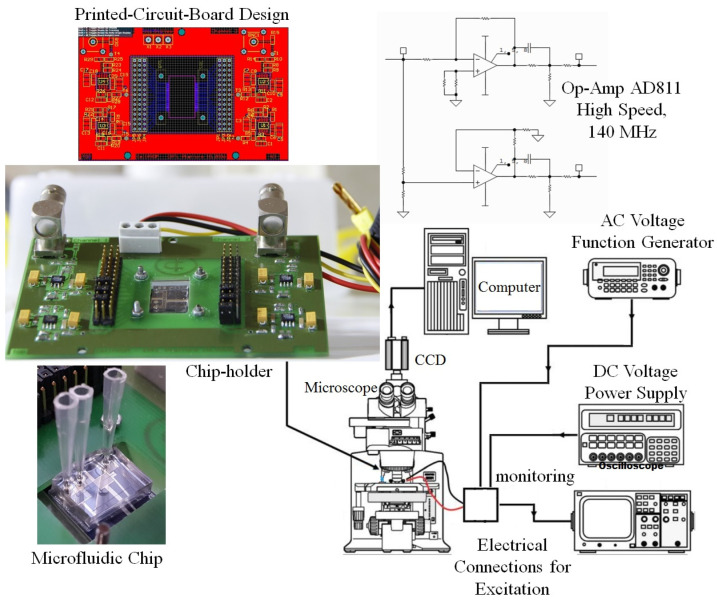
Experimental setup for micromanipulator test.

**Figure 11 biomimetics-10-00056-f011:**
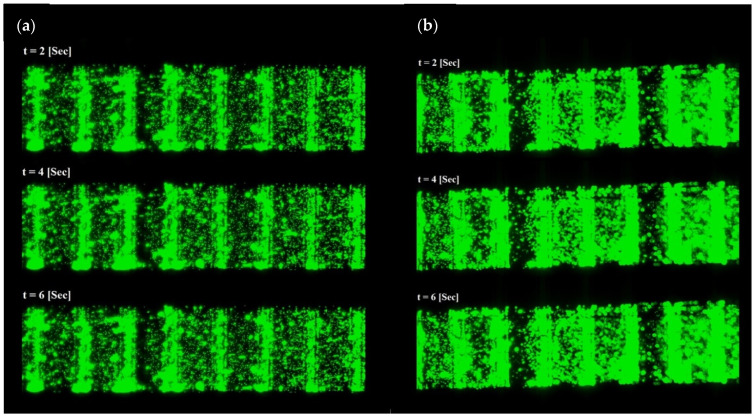
Particles motion results due to generated ACET force: (**a**) 0.2 S/m, +/−3 V_rms_, and 300 kHz; (**b**) 0.6 S/m, +/−3 V_rms_, and 300 kHz.

**Figure 12 biomimetics-10-00056-f012:**
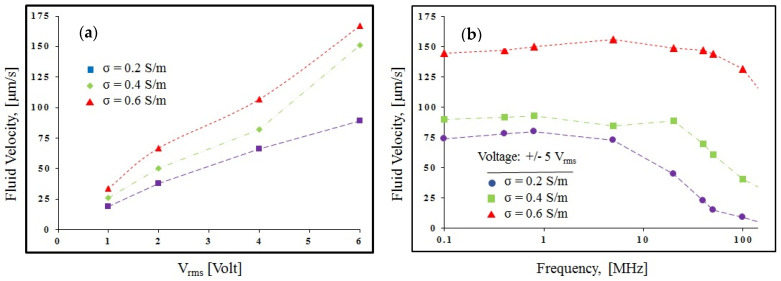
Experimental results for fluid velocity: (**a**) electric potential and σ effect; (**b**) frequency effect (from 100 kHz to 140 MHz).

**Figure 13 biomimetics-10-00056-f013:**
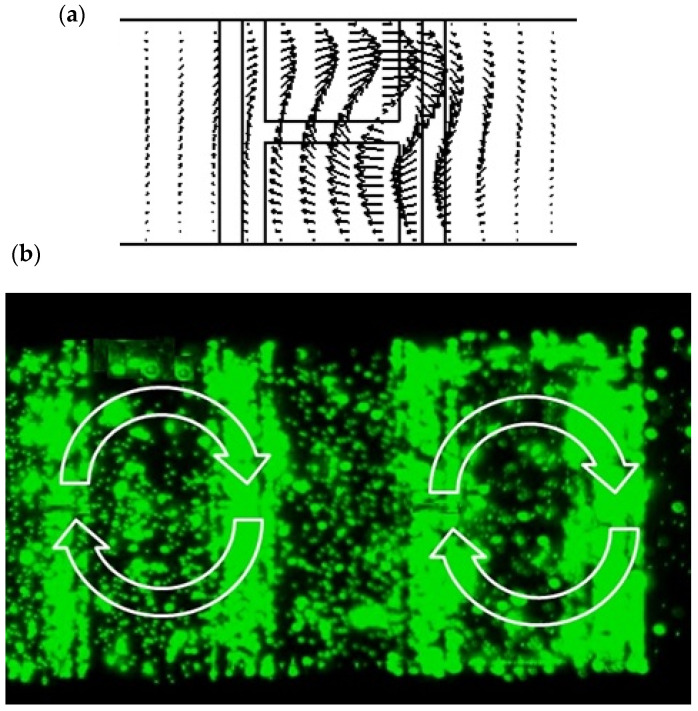
AC electrothermally driven rotational flow: (**a**) finite element analysis results; (**b**) experimental results.

**Table 1 biomimetics-10-00056-t001:** Fluid material properties.

Symbol	Description	Value
*D*	Diffusion coefficient	2 × 10^−11^ m^2^.s^−1^
*U*	Average inlet velocity	0.1 mm.s^−1^
*ρ_m_*	Viscosity of fluid	1 × 10^−3^ N.s.m^−2^
*μ*	Density of fluid	10^3^ kg.m^−3^
*σ*	Electric conductivity of fluid	0.2–0.6 [S/m]
*ε_r_*	Dielectric constant of fluid	80.2
*ε_0_*	Vacuum permittivity	8.854 × 10^−12^ F.m^−1^
*f*	Applied electric field frequency	100 kHz–140 MHz
*V_0_*	Electric potential	1–8 V
c_p_	Heat capacity of fluid	4.184 [kJ/(kg × K)]
k	Thermal conductivity of fluid	0.598 [W/(m × K)]
T_ambient_	Ambient temperature	300.15 [K]
C_in1_	Species concentration of fluid A	1 [mol.m^−3^]
C_in2_	Species concentration of fluid B	0 [mol.m^−3^]

## Data Availability

The raw data supporting the conclusions of this article will be made available by the authors on request.

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
