# Peer review of "Theoretical and Experimental Study of an Electrokinetic Micromanipulator for Biological Applications"

_biomimetics, 2025, doi:10.3390/biomimetics10010056_

Round 1
Reviewer 1 Report
Comments and Suggestions for Authors
The paper discusses the design, simulation, fabrication, and testing of a microfluidic chip that uses AC electrothermal flow for manipulating biological fluids within microchannels. This technology is useful for applications in biological samples . However, the core of this technology should be more explained in comparison with other alternatives.
In the Introduction, writing style needs to be improved. The author refer to references numbers as if it is a review articles. Please write in concise way (40~90).
The explanation of the fluid dynamics can be expanded to provide a clearer understanding of how the AC electrothermal forces alter the fluid flow within the microchannels. Visual materials (video) or more detailed diagrams showing the flow dynamics and electrode interactions could enhance comprehension.
Further thermal analysis might be needed, as the generation of heat within microchannels could affect biological samples. Or at least that should be discussed in the discussion.
The bio-compatibility of the materials used and their effects on sample integrity during experiments will be important, especially for biological applications
Figure 8. The fluorescence signal from particles is almost saturated. Better representation of data is recommended.
Author Response
Reviewer#1:
Comments:
The paper discusses the design, simulation, fabrication, and testing of a microfluidic chip that uses AC electrothermal flow for manipulating biological fluids within microchannels. This technology is useful for applications in biological samples. However, the core of this technology should be more explained in comparison with other alternatives.
- While thanking the esteemed reviewer for a thorough review of the manuscript version. We, the authors of the article, believe that your suggestions have been very useful and effective in improving the scientific version of the manuscript. We carefully answered all the questions and suggestions of the esteemed reviewer and added them to the manuscript version.
- In the Introduction, writing style needs to be improved. The author refer to references numbers as if it is a review articles. Please write in concise way (40~90).
- By considering the comments, we read the manuscript again and improve the writing quality. Please check the green highlights. In introduction part. For example page2 lines (54,56, 66-69, 92-94, 103, 104, 107, 109 and etc);
- The explanation of the fluid dynamics can be expanded to provide a clearer understanding of how the AC electrothermal forces alter the fluid flow within the microchannels. Visual materials (video) or more detailed diagrams showing the flow dynamics and electrode interactions could enhance comprehension.
- Some explanations are added in introduction part and also some new figures are added to the manuscript (section 1 and 3). The manuscript is revised based on this comment. Please check Page2, lines (74-90); page7 lines (202-206); page 9 Figure.4; Page 10 lines (249-255); Page 11 figure.6
As shown in Figure.1; by placing two pairs of electrodes at the bottom of a microchannel and applying a high-frequency AC signal, three different frequency range can be discussed:
- a) when the external ac electric field is more slower than the fluid charging time (τq = ε/σ); since there is enough time to fully screen the electrodes by the electric field, therefore no leaving electric field in the medium.
- b) When the external ac electric field is comparable to the fluid charging time (τq = ε/σ) which shown in Figure. 1a; the electrodes are partially screened by a part of applied electric field and the other part of electric field falls over the fluid medium, as a result the corresponding tangential force on the ions will cause AC electroosmotic flow in the medium.
- c) when the external ac electric field is more faster than the fluid charging time (τq = ε/σ) as illustrated in Figure. 1b; since there is no time to screening the electrodes, therefore all the electric field falls over the fluid medium but the ions have not any role in the formation of bulk electric field. In this frequency range, the ACEO flow is very poor and instead, the ACET mechanism can be occurred due to the thermal effects inside the channel [15].
Figure. 1 ac electric field and fluid interaction at different frequencies; a) applied electric field frequcy is comparible with the fluid frequency, b) applied electric field frequcy is much higher than the fluid frequency.
A temperature rise will be induced by applying an electric potential over the ionic strength mediums such as biological solutions. The generated temperature increment can be estimated by heat balance equation (Lian et al., 2007):
(1)
In which U is the velocity vector, T is the temperature, cp is the specific heat, ρ is the fluid density, k is the thermal conductivity of medium. In addition, by applying a non-uniform electric field a temperature gradients field will be produce inside the microchannel. The generated temperature gradients cause gradients in electrical properties (σ and ε which represent electrical conductivity and electrical permittivity, respectively). The dependence of temperature on the electrical properties of fluid can be estimated by Eq. 2 and Eq. 3 (Haynes and Lide, 2010):
(2)
(3)
Where T0 is the ambient temperature. The time-averaged ACET force per unit volume can be written as (Lian et al., 2007):
(4)
Where ω=2πfc is the angular frequency of ac electric field and τ=ε(T0)/σ(T0) is the charge relaxation time of medium. The induced ac electrothermal flow FET, causes bulky fluid motion to rise inside the channel.
It should be noted that the results in Figure. 3 are simulated by using a unit cell with Periodic Boundary Condition (PBC). PBC means a set of boundary conditions which are chosen for approximating a large (infinite) system by using a small part called a unit cell. For more visualization we simulate the system with four units of electrode pairs, and as shown in Figure.4 same results are achieved.
Figure 4. simulation results for four units of electrodes and resulting the generated AC electrothermal flow.
Also, in order to more investigate the chaos mixing effect we released two adjacent particles at the inlet of microchannel. As shown in Figure.6 the particle trajectories illustrate that the adjacent particles will experience a bulk force and as a result the particles are stretched, folded and finally separated (a proof of chaotic regime) [11, 42]. The stretching, folding and breaking up effects reveals chaotic regime and eases the molecular diffusion transport for mixing purposes.
Figure. 6 Particle trajectories and the resulted stretching and folding effects.
- Further thermal analysis might be needed, as the generation of heat within microchannels could affect biological samples. Or at least that should be discussed in the discussion.
- The maximum temperature rise limits the applications of the proposed AC electrothermally-driven microchip. Based on the references maximum temperature rise of 5 ï‚°C is acceptable for most biological applications. The manuscript is revised based on this comment. Please check pages 9 and 10 lines (235-246).
High temperature rise induce mass density difference in fluid and as a result, buoyancy effect arises inside the channel. the ratio of ACET force over the buoyancy force investigated by Eq. (16) [38].
(16) |
|
where g = 9.8 m/s2 is the gravity acceleration, ∂ρm/∂T, accounts for thermal expansion of fluid and L is the characteristic length of the system. It was proven that the ACET force is much stronger than buoyancy force, when the temperature rise is about 5 K or less [38]. In addition to the buoyancy effect, such large temperature rises can damage solution in bio-analytical and immunoassay binding applications [41]. We generate an efficient motional effect by locally small temperature distribution inside the channel. It can be seen from the results that the maximum temperature rise for different applications can be controlled based on the applied electric potential and the fluid ionic strength.
- The bio-compatibility of the materials used and their effects on sample integrity during experiments will be important, especially for biological applications
- The PDMS microchannel and Pt electrodes are super biocompatible materials for most biological samples. The manuscript is revised based on this comment. Please check page 14 lines (343-349)
One of the most important requirements for biofluidic chips are biocompatibility, the PDMS microchannel and Pt electrodes are well known biocompatible materials for most biological samples, DNA analysis, PCR, enzyme assays and integrated analytical detections such as on chip NMR, FRIT and Raman spectroscopy [44-48]. It should be noted the parylene coating can improve the biocompatibility of a microfluidic device [49].
- Figure 8. The fluorescence signal from particles is almost saturated. Better representation of data is recommended.
- Figure 11 a illustrates the movement paths of particles over time, the average speed in this test was 70 μm/s. while Figure 11b indicates average fluid velocity 145 μm/s for 0.6 [S/m] solution. Given the prevailing conditions, this was the best condition for fluorescent display.

Reviewer 2 Report
Comments and Suggestions for Authors
Microfluidics is currently one of the popular technologies for analyzing a variety of objects of different structure and nature. The article by Vafaie et al. is devoted to the design and manufacture of such a device. The article is of interest, but requires a number of clarifications:
1. The title of the article does not reflect its content. It is important to indicate that an electrokinetic study is being conducted.
2. In the Introduction, it would be useful for the authors to note the key advantages of the electrokinetic mixing principle with arguments from the literature.
3. Explanations of the formulas require the addition of units of measurement, which is important for readers to understand the physical nature of the phenomenon described.
4. Table 1 presents the data on the studied liquid. Why did not the authors study the effect of its various parameters on flow control? The chemical composition and ionic strength of the liquid greatly affect the possibility of conducting any electrophysical experiments in it.
4. The technology of forming microfluidic channels is well known and does not represent serious scientific interest. At the same time, in Figure 6, the authors presented a whole multitude of electrode variants without presenting even a long simulation of the influence of these topologies on the experimental parameters. In Figure 6, there is no explanation of what each element on the substrate represents.
5. The experimental results presented in Figure 9 require statistical evaluation.
6. In conclusion, it is necessary to add a discussion of the potential use of the developed device.
Comments on the Quality of English LanguageEnglish needs improvement for better presentation and understanding of data.
Author Response
Reviewer#2:
Comments:
Microfluidics is currently one of the popular technologies for analyzing a variety of objects of different structure and nature. The article by Vafaie et al. is devoted to the design and manufacture of such a device. The article is of interest, but requires a number of clarifications:
- While thanking the esteemed reviewer for a thorough review of the manuscript version. We, the authors of the article, believe that your suggestions have been very useful and effective in improving the scientific version of the manuscript. We carefully answered all the questions and suggestions of the esteemed reviewer and added them to the manuscript version.
- The title of the article does not reflect its content. It is important to indicate that an electrokinetic study is being conducted.
- The manuscript is revised based on this comment and the title changed to “Theoretical and experimental study of an electrokinetic micromanipulator for biological applications”. Please check page1 lines (2 and 3)
- In the Introduction, it would be useful for the authors to note the key advantages of the electrokinetic mixing principle with arguments from the literature.
- The manuscript is revised based on this comment. Please check page 3 lines (110-117)
Alternating positive and negative sinusoidal signals were applied to the electrodes to form electric and thermal effects that generate flows within the channel. In clinical medicine and biological test, fluidic micro-manipulators have been widely employed to identification of biochemical products, enzyme assay, polymerization, organic synthesis and biological screening applications [19-22]. Using electrokinetic effect for mixing applications has many advantages including miniaturization, no moving components, simple design, fast and efficient mixing, negligible hydrodynamic dispersion, no vibration and fatigue and easier integration with microelectronics [23-25].
- Explanations of the formulas require the addition of units of measurement, which is important for readers to understand the physical nature of the phenomenon described.
- Yes, the respected referee's opinion is absolutely correct. However, we have stated the units of measurement of the formulas in the manuscript.
- Table 1 presents the data on the studied liquid. Why did not the authors study the effect of its various parameters on flow control? The chemical composition and ionic strength of the liquid greatly affect the possibility of conducting any electrophysical experiments in it.
- The effect of impressive parameters such as electric signal amplitude, electric signal frequency and the fluid electric conductivity (ionic strength) is studied based on Figures 7 and 12a,b. It should be noted that we also used Taguchi optimization method for our geometrical design. The electrode size and the channel geometrical optimization was not the main objective of our research; that is why we are not interested in adding this part inside the manuscript. Based on your comment, the name of optimization method is mentioned on the manuscript. Please check page 13 line 94.
We have already optimized our geometrical design by well-known Taguchi method [Taguchi G, Chowdhury S, Wu Y. Taguchi’s quality engineering handbook. John Wiley & Sons; 2004.].
For your information, our optimization details are described here:
Taguchi developed a special design of orthogonal arrays to study the entire parameter space with a small number of experiments only. The experimental results are then transformed into a signal-to-noise (S/N) ratio. It uses the S/N ratio as a measure of quality characteristics deviating from or nearing to the desired values. Taguchi suggested the use of the following SN ratios:
For the-smaller-the-better characteristic,
Eq-a |
Where y is the response, and n is the number of tests in a trial. For the-larger-the-better characteristic,
Eq-b |
In practice, the performance of the-smaller-the-better and the-larger-the-better characteristic problems is judging by the SN ratio. The higher the SN ratio, the better performance is. The merits and shortcomings of the Taguchi methods can be found in [Taguchi G, Chowdhury S, Wu Y. Taguchi’s quality engineering handbook. John Wiley & Sons; 2004.]. We propose below steps for electrode spacing and geometrical parameters optimization for ACET microchip:
Step 1: Identification of control factors, and their levels,
Step 2: Experiment design; the orthogonal array is determined and control factors are assigned to the selected orthogonal array.
Step 3: Coupled multi-physic FEM Simulation and data collection.
Step 4: Data analysis.
In our research, six parameters are adjustable and considered potentially influential in improving ACET velocity effect. The control factors are substrate material and geometrical dimensions (Table. A, Figure. A). By collecting several trial data on parameter setting and the corresponding ACET motion, we determined control factor levels based on engineering knowledge and experience. This study considered one control factors with two levels and five control factors with four levels. The mixed-level orthogonal array of L32 (21×45) was employed by Taguchi method to verify the effects of these factors on pumping performance (Table. B). For each trial, independent multi-physic FEM simulation were done.
Table A. Factors and their levels.
Factors |
Unit |
Levels |
|||
1 |
2 |
3 |
4 |
||
Substrate Material |
- |
Si |
Glass |
- |
- |
microChannel Width |
µm |
100 |
200 |
300 |
400 |
N.E |
µm |
2.5 |
5 |
7.5 |
10 |
G |
µm |
2.5 |
5 |
7.5 |
10 |
C.E |
µm |
10 |
20 |
30 |
40 |
G/2 |
µm |
30 |
40 |
50 |
60 |
Figure. A geometrical dimension for ACET microchip
Table B. A mixed-level ortogonal array L32 (21×45).
Trial Number |
Factors and their levels |
Results |
|||||
Substrate |
Width |
N.E |
G |
C.E |
L-G |
Velocity (μm/sec) |
|
1 |
1 |
1 |
1 |
1 |
1 |
1 |
62 |
2 |
1 |
2 |
2 |
2 |
2 |
2 |
68 |
3 |
1 |
1 |
3 |
3 |
3 |
3 |
60 |
4 |
1 |
1 |
4 |
4 |
4 |
4 |
49 |
5 |
1 |
2 |
1 |
1 |
2 |
2 |
65 |
6 |
1 |
2 |
2 |
2 |
1 |
1 |
63 |
7 |
1 |
2 |
3 |
3 |
4 |
4 |
55 |
8 |
1 |
2 |
4 |
4 |
3 |
3 |
53 |
9 |
1 |
3 |
1 |
2 |
3 |
4 |
59 |
10 |
1 |
3 |
2 |
1 |
4 |
3 |
59 |
11 |
1 |
3 |
3 |
4 |
1 |
2 |
58 |
12 |
1 |
3 |
4 |
3 |
2 |
1 |
57 |
13 |
1 |
4 |
1 |
2 |
4 |
3 |
58 |
14 |
1 |
4 |
2 |
1 |
3 |
4 |
56 |
15 |
1 |
4 |
3 |
4 |
2 |
1 |
56 |
16 |
1 |
4 |
4 |
3 |
1 |
2 |
57 |
17 |
2 |
1 |
1 |
4 |
1 |
4 |
48 |
18 |
2 |
1 |
2 |
3 |
2 |
3 |
58 |
19 |
2 |
1 |
3 |
2 |
3 |
2 |
58 |
20 |
2 |
1 |
4 |
1 |
4 |
1 |
49 |
21 |
2 |
2 |
1 |
4 |
2 |
3 |
52 |
22 |
2 |
2 |
2 |
3 |
1 |
4 |
47 |
23 |
2 |
2 |
3 |
2 |
4 |
1 |
53 |
24 |
2 |
2 |
4 |
1 |
3 |
2 |
51 |
25 |
2 |
3 |
1 |
3 |
3 |
1 |
51 |
26 |
2 |
3 |
2 |
4 |
4 |
2 |
50 |
27 |
2 |
3 |
3 |
1 |
1 |
3 |
52 |
28 |
2 |
3 |
4 |
2 |
2 |
4 |
53 |
29 |
2 |
4 |
1 |
3 |
4 |
2 |
49 |
30 |
2 |
4 |
2 |
4 |
3 |
1 |
49 |
31 |
2 |
4 |
3 |
1 |
2 |
4 |
48 |
32 |
2 |
4 |
4 |
2 |
1 |
3 |
49 |
SN ratios were calculated by Eq-b for each trial. Thereafter, the response graphs of each factor to the SN ratio were drawn (Fig. B).
Fig. B Main factor effects on ACET pumping effect.
Based on the analyses, the SNR curves show that the optimal values can be set in according with Table C.
Table C. Geometrical dimensions of design.
Symbol |
Description |
Value (μm) |
WChannel |
Microchannel width |
200 |
L.E |
Large Electrode |
20 |
N.E |
Narrow Electrode |
5 |
C.E |
Center Electrode |
20 |
g |
Gap between two electrodes |
5 |
G |
Gap between two pairs of electrode structures |
80 |
- The technology of forming microfluidic channels is well known and does not represent serious scientific interest. At the same time, in Figure 6, the authors presented a whole multitude of electrode variants without presenting even a long simulation of the influence of these topologies on the experimental parameters. In Figure 6, there is no explanation of what each element on the substrate represents.
- As discussed in previous section different geometrical sizes are designed for ACET the proposed microchip. The efficient sizes are selected based on Taguchi optimization results. The manuscript is revised based on this comment. Please check page 13 lines (292-297)
It should be noted that different geometrical sizes are designed for ACET the proposed microchip. The efficient sizes are selected based on Taguchi optimization results. Therefore 16 different topologies for electrode structures are fabricated for different applications such as pumping, mixing and concentrating. The electrical connections of the microchips are tested and the chips are prepared for experimental investigation.
- The experimental results presented in Figure 9 require statistical evaluation.
- According to the respected reviewer, in articles related to macrofluidic devices, the relationship between velocity and various parameters is shown in Figure 5, and none of the recent references have examined statistical analysis for this relationship.
- In conclusion, it is necessary to add a discussion of the potential use of the developed device.
- Yes, the opinion of the respected referee is completely correct, the manuscript is revised based on this comment. Please check page 18 lines (427-431)
Integration of conventional detection/analytical methods (such as NMR, IR transmission, FTIR and Raman spectroscopy) with our developed microchip will be of interest for variety of applications such as molecular structure analyzes [44, 45], enzyme assay [46, 47] and protein folding/refolding [48, 52] applications.
References:
- Liu RH, Yang J, Lenigk R, Bonanno J, Grodzinski P. Self-contained, fully integrated biochip for sample preparation, polymerase chain reaction amplification, and DNA microarray detection. Analytical chemistry. 2004 Apr 1;76(7):1824-31.
- Yun KS, Yoon E. Microfluidic components and bio-reactors for miniaturized bio-chip applications. Biotechnology and Bioprocess Engineering. 2004 Apr;9:86-92.
- Chen X, Cui D, Liu C, Li H, Chen J. Continuous flow microfluidic device for cell separation, cell lysis and DNA purification. Analytica chimica acta. 2007 Feb 19;584(2):237-43.
- Ukita Y, Asano T, Fujiwara K, Matsui K, Takeo M, Negoro S, Kanie T, Katayama M, Utsumi Y. Application of vertical microreactor stack with polystylene microbeads to immunoassay. Sensors and Actuators A: Physical. 2008 Jul
- Lavrentovich OD. Transport of particles in liquid crystals. Soft Matter. 2014;10(9):1264-83.
- Garrido J, Ibanez JA, Pellicer J, Tejerina AF. On the streaming current of electrokinetic processes. Journal of Electrostatics. 1982 Apr 1;12:469-76.
- Jayaram S, Cross JD, Weckman EJ. Electrokinetic study in non-polar liquids by laser Doppler anemometry. Journal of electrostatics. 1995 Feb 1;34(1):1-6.
- Vafaie RH, Mehdipour M, Pourmand A, Ghavifekr HB. A novel miniaturized electroosmotically-driven micromixer modified by surface channel technology. In20th Iranian Conference on Electrical Engineering (ICEE2012) 2012 May 15 (pp. 124-129). IEEE.
- Petkewich R. Research Profiles: Integrating micromixer and microcoils for time-resolved NMR.
- Tseng WK, Lin JL, Sung WC, Chen SH, Lee GB. Active micro-mixers using surface acoustic waves on Y-cut 128 LiNbO3. Journal of Micromechanics and Microengineering. 2006 Feb 6;16(3):539.
- Miller EM, Wheeler AR. A digital microfluidic approach to homogeneous enzyme assays. Analytical Chemistry. 2008 Mar 1;80(5):1614-9.
- Dittrich PS, Manz A. Lab-on-a-chip: microfluidics in drug discovery. Nature reviews Drug discovery. 2006 Mar 1;5(3):210-8.
- Kakuta M, Hinsmann P, Manz A, Lendl B. Time-resolved Fourier transform infrared spectrometry using a microfabricated continuous flow mixer: application to protein conformation study using the example of ubiquitin. Lab on a Chip. 2003;3(2):82-5.
- Nguyen NT, Wereley ST, Shaegh SA. Fundamentals and applications of microfluidics. Artech house; 2019 Jan 31.
- Chan CK, Hu Y, Takahashi S, Rousseau DL, Eaton WA, Hofrichter J. Submillisecond protein folding kinetics studied by ultrarapid mixing. Proceedings of the National Academy of Sciences. 1997 Mar 4;94(5):1779-84.

Round 2
Reviewer 1 Report
Comments and Suggestions for Authors
Authors did responded to my comments efficiently
Reviewer 2 Report
Comments and Suggestions for Authors
The authors have substantially revised the article. It may be published in its current form.